# Sustainable Design of Innovative Kiwi Byproducts-Based Ingredients Containing Probiotics

**DOI:** 10.3390/foods11152334

**Published:** 2022-08-05

**Authors:** Gheorghe-Ionuț Ilie, Ștefania-Adelina Milea, Gabriela Râpeanu, Adrian Cîrciumaru, Nicoleta Stănciuc

**Affiliations:** 1Faculty of Food Science and Engineering, Dunarea de Jos University of Galati, 800201 Galați, Romania; 2Cross-Border Faculty, Dunarea de Jos University of Galati, 800201 Galați, Romania

**Keywords:** kiwi byproducts, probiotic, prebiotic, antioxidant activity, *Lactobacillus casei*, ingredients, functional foods

## Abstract

Industrial processing of kiwifruits generates a large quantity of byproducts, estimated to be one million tons per year. The resulting byproducts are rich sources of bioactive components that may be used as additives, hence minimizing economic and environmental issues. In this study, kiwifruit byproducts were used to develop added-value food-grade ingredients containing probiotics. The byproducts were divided into peels and pomace. Both residues were inoculated with a selected strain of probiotic (*Lacticaseibacillus casei* 431^®^), and two variants were additionally enhanced with prebiotic sources (buckwheat and black rice flours). The inoculated powders were obtained by freeze-drying, and the final ingredients were coded as KP (freeze-dried kiwi peels), KBR (freeze-dried kiwi pomace and black rice flour), KPB (freeze-dried kiwi pomace and buckwheat flour), and KPO (freeze-dried kiwi pomace). The phytochemical profile was assessed using different spectrophotometric methods, such as the determination of polyphenols, flavonoids, and carotenoids. The kiwi byproduct-based formulations showed a polyphenolic content varying from 10.56 ± 0.30 mg AGE/g DW to 13.16 ± 0.33 mg AGE/g, and the survival rate of lactic acid bacteria after freeze-drying ranged from 73% to 88%. The results showed an increase in total flavonoid content from the oral to gastric environment and controlled release in the intestinal environment, whereas a maximum survival rate of probiotics at the intestinal end stage was 48%. The results of SEM and droplet size measurements revealed vesicular and polyhedral structures on curved surfaces linked by ridge sections. The CIEL*a*b* color data were strongly associated with the particular pigment in kiwi pulp, as well as the color of the additional flour. Finally, the ingredients were tested in protein bars and enhanced the value of the final food product regarding its phytochemical and probiotic content.

## 1. Introduction

The industrial processing of fruits and vegetables leads to large quantities of byproducts, such as peels, seeds, and residual pulp, which are usually used as animal feed or/and discarded in the environment. Nowadays, enormous efforts, from both scientific and application points of view, are made to explore the functional potential of these byproducts as sources of important chemical compounds with significant added value for human and animal health. In general, these compounds are represented by polyphenols, carotenoids, and triterpenes [1], which display several biological activities, namely, antioxidant [2], anti-inflammatory [3], antimicrobial [4], and antidiabetic effects [5], among many others [6].

The genus *Actinidia* contains about 60 species, with kiwi probably being the most consumed fruit in natura due to its well-described health benefits [7]. Kiwifruits can be preserved in the fresh state for a prolonged time (for many months) without any decrease in quality, requiring controlled temperature (0 °C). At the industrial level, innovative kiwi products are processed and commercialized, such as juice, frozen juice, sweets, and ice creams, among other products [8]. Industrial processing of kiwifruits causes significant amounts of peels and pomace, which are still under-explored and have aroused great interest due to their high contents of bioactive molecules, such as phenolic compounds [9]. The development of kiwi-derived products is based on their nutritional and biological properties due to their rich contents of dietary fiber and bioactive compounds, such as vitamins (C, E, and A), phenolic compounds, and minerals [6].

Significant data from the literature have shown that one of the most recommended ways to prevent and treat gastrointestinal alterations is to alter the commensal microbiota [10,11,12]. It is well known that microorganisms coexist with eukaryotic cells at the mucosal surfaces of vertebrates in a complex and harmonious symbiosis [11]. However, the administration of antibiotics at a large scale causes various side effects, such as disrupting normal microflora in the human body by destroying normal gut and genital tract bacteria [13]. These imbalances can be adjusted by probiotic supplementation in the diet, providing a load of live bacterial cells sufficient for beneficial effects on human health at both metabolic and immune levels [11]. The engineering of probiotic foods is challenging due to their low acid-bile tolerance and requirements for their non-pathogenicity, nontoxicity, ability to survive and metabolize in the gut environment, resistance to low pH and organic acid, and potential to remain viable for a long period under storage and other conditions [14]. An optimal population of probiotic bacteria is critical for the preservation and proper functioning of the digestive system [15].

Prebiotics are defined as non-fermentable components that are transferred into the colon to be selectively used by host microorganisms [16]. Multiple benefits have been suggested for prebiotics, such as the mediation of host health (such as improved intestinal function), the regulation of glucose and lipid metabolism, immune response, bone health, and the regulation of satiety [17]. It has been suggested that prebiotics can be classified into three categories: oligosaccharides (fructooligosaccharide, xylooligosaccharide, galactooligosaccharide, isomaltose, inulin, etc.), fiber (β-glucan, pectin, cellulose, dextrin, etc.), and polyols (xylitol, mannitol, lactulose, etc.) [17]. Other compounds have been proposed as candidates for prebiotics, such as linoleic acid, polyunsaturated fatty acids, phenols, and polyphenols, such as anthocyanins [18].

Given the need to use prebiotics in the engineering of probiotic foods, different complex food matrices should be considered, including fibers, polyphenolics, and oligosaccharides. For example, black rice (*Oryza sativa* L.) flour is a valuable source of protein, fat, carbohydrates, phenols, flavonoids, and anthocyanins [19] and could be a potent candidate for complex prebiotic activity. Additionally, buckwheat is a type of underutilized pseudocereal belonging to the genus *Fagopyrum*, and it could be regarded as a potential source for food and nutritional applications [20]. According to FoodData Central [21], the proximate composition of buckwheat includes starch as the major component (~70%), followed by protein (~12%), dietary fiber (~10%), lipids (~3%), and ash (2.5%); minor components with biological significance, such as polyphenols, D-chiro-inositol, and vitamins, were also reported [22].

Therefore, our study aims to contribute to the innovative development of kiwi byproduct-based products containing probiotics. Although the kiwi phytochemical profile is already advanced in the scientific literature, based on our knowledge, no studies have explored the potential of transforming kiwi byproducts into foods and ingredients containing probiotics, consequently reducing food waste. Kiwi pomace was enhanced with two prebiotic sources, namely, black rice and buckwheat flour, and inoculated with *Lacticaseibacillus casei* (*L. casei*) 431^®^, whereas kiwi peels were inoculated to the same extent with *L. casei*. Four powders containing probiotics were obtained by freeze-drying and characterized for phytochemicals (anthocyanins, polyphenols, and carotenoids), antioxidant activity, color, cell viability, and the bioaccessibility of polyphenols and probiotics. The powders were tested for food applications by introducing them into a protein bar formulation. The foods were tested for phytochemicals, antioxidant activity, and cell viability to assess their potential as functional foods.

## 2. Materials and Methods

### 2.1. Chemicals

Analytical-grade n-hexane, acetone, 2,2-diphenyl-1-picrylhydrazyl (DPPH), 6-hydroxy-2,5,7,8-tetramethylchromane-2-carboxylic acid (Trolox), acetic acid, hydrochloric acid, aluminum chloride, ethanol, Folin–Ciocâlteu reagent, gallic acid, pectin, and methanol were purchased from Sigma Aldrich Steinheim (Darmstadt, Germany). Other reagents such as sodium bicarbonate were purchased from Honeywell, Fluka (Seelze, Germany), and *Lacticaseibacillus casei (L. casei)* 431^®^ strain was purchased from Chr. Hansen (Hoersholm, Denmark). The probiotic character of *L. casei* 431^®^ strain has been described in several studies [23,24] and has been associated with health benefits in several areas of health, including immune health, respiratory health, and bowel function. De Man, Rogosa, and Sharpe agar (MRS agar) was purchased from Merck (Darmstadt, Germany). All reagents and solvents were of analytical and HPLC grade.

### 2.2. Fruit Processing

Kiwifruits (*Actinidia deliciosa* cv. “Hayward”) at full maturity were purchased in November 2021 from a local supermarket in Galati County, Romania. The fruits were sorted and processed immediately. Once they arrived at the laboratory, the fruits were washed with distilled water, and peels were separated from the fruits. The fruits were further cut in half and squeezed using a juice extractor (Stainless Steel Fruit Vegetable Juice Extractor Juicer Squeezer, Guangdong, China). The fresh material, specifically peels and pomace, was used for initial phytochemical content and antioxidant activity analyses. 

### 2.3. Extraction of Phytochemicals from Fresh Kiwi Samples

In order to characterize the freeze-dried powders of the fresh material, a subsequent extraction was performed using a solid–liquid ultrasound-assisted method with two different solvents. The design of extraction procedures was based on a comprehensive comparison of both lipophilic and hydrophilic profiles by using two different combinations of solvents: a mixture of ethanol and water (ratio of 70:30, *v*/*v*) and *n*-hexane–acetone (ratio 1:3, *v*/*v*). The fresh material was extracted with the corresponding solvent solutions in a solid–liquid ratio of 1:10 by mixing and ultrasound treatment at 35 °C for 30 min. Each extraction was repeated twice to obtain enriched extracts. 

### 2.4. Customized Design to Develop Kiwi-Based Ingredients Enriched with Probiotics

About 50 g of fresh material (kiwi peels and pomace) was added to ultrapure water in a ratio of 1:2 and mixed by homogenization (ProBlend Crush Blender, Global Headquarters Netherlands, Eindhoven). Prior to inoculation, the mixtures were enhanced with different flours, i.e., black rice and buckwheat (20%), and allowed to hydrate, followed by pH adjustment to 5.0 with 1.0 N NaOH. The designed samples were coded as follows: KP (freeze-dried kiwi peels), KBR (freeze-dried kiwi pomace and black rice flour), KPB (freeze-dried kiwi pomace and buckwheat flour), and KPO (freeze-dried kiwi pomace). All samples were sterilized using a UV lamp and inoculated with 2% *L. casei* 431^®^ and freeze-dried (CHRIST Alpha 1–4 LD plus, Osterode am Harz Germany) at −42 °C under a pressure of 10 Pa for 48 h. Afterwards, the powders were collected and packed in glass containers and kept at 4 °C until further analysis.

### 2.5. Total Polyphenol (TPC) and Total Flavonoid (TFC) Analysis

Spectrophotometric methods were used for the TPC and TFC evaluation by using the Folin–Ciocâlteu reagent and aluminum chloride methods, respectively, as described by Milea et al. [25]. For TP content, from each sample extract, a volume of 0.2 mL was diluted with 15.8 mL of distilled water, followed by the addition of 1 mL of Folin–Ciocâlteu reagent and 3 mL of 20% Na_2_CO_3_. The mixtures were left to stand for one hour in the dark to react, followed by reading the absorbance at λ = 765 nm (Jenway Scientific Instruments, Essex, UK). TP was expressed as mg gallic acid equivalents/g dry weight (mg GAE/g DW) using a calibration curve. For TF evaluation, a volume of 0.25 mL from each extract was sequentially mixed with 0.075 mL of sodium nitrite (5%), 0.15 mL of 10% AlCl_3_, and 0.5 mL of 1 M NaOH. The absorbance of the mixtures was immediately measured at 510 nm against the suitable blank. TF was expressed in mg catechin equivalents (CE) per g of dry powder (mg CE/g DW). 

### 2.6. Total Monomeric Anthocyanin Content 

Due to the special formulation of the KBR sample, which contained freeze-dried kiwi pomace and black rice flour, the total monomeric anthocyanin content (TAC) was assessed using the official AOAC method [26] and expressed as mg cianidin-3-O-glucoside equivalents/g DW (mg C3G/g DW).

### 2.7. Carotenoid Content Evaluation

Carotenoid content, specifically β-carotene and lycopene, was determined by the spectrophotometric method by selecting different wavelengths: 470 nm (β-carotene) and 503 nm (lycopene). The amount of carotenoids was calculated according to the following equation [27]: (1)$$Carotenoids\;\left(\frac{mg}{g}\right)=\frac{A\times M_{w}\times D_{f}}{M_{a}\times L}$$
where *A* is the absorbance of the extracts at the corresponding wavelength; *M_w_* is molecular weight, *D_f_* is the sample dilution rate, *M_a_* is molar absorptivity (2500 L mol^−1^ cm^−1^ and 3450 L mol^−1^ cm^−1^, respectively), and *L* is the cell diameter of the spectrophotometer (1 cm).

### 2.8. Radical Scavenging Activity 

In order to evaluate the radical scavenging activity of the extracts, two methods were comparatively used, namely, the scavenging percentages of DPPH and ABTS radicals [28]. In order to evaluate the DPPH radical scavenging activity, a volume of 0.1 mL from the hydrophilic extract was mixed with 3.9 mL of DPPH solution (1 mM in methanol), mixed, and allowed to react for 1 h at room temperature in the dark. For ABTS radical scavenging activity, a volume of 0.15 mL from the lipophilic extract was mixed with a volume of 2.9 mL ABTS solution (7 mM ABTS in 2.45 mM K_2_S_2_O_8_) and allowed to react for 2 h at room temperature in the dark. The radical scavenging activity was expressed as mMol Trolox/g DW using a calibration curve.

### 2.9. Viable Counts of L. casei 431^®^

In order to evaluate the viable counts of *L. casei* 431^®^, 10-fold serial dilutions of the freeze-dried powders were performed using sterile physiological serum (0.9 NaCl %, *w*/*v*) by using the pour plate technique, as described by Vasile et al. [29]. The viable cell number was determined by estimating the number of colony-forming units (CFU) by cultivation on MRS agar plates (medium at pH 5.7) after 72 h of aerobic incubation at 37 °C. The counts were expressed as CFU/g DW.

### 2.10. In Vitro Digestion 

In order to evaluate the bioaccessibility of the bioactives and probiotics from powders, the KPB sample was selected in order to evaluate the flavonoids and probiotic viability in a simulated in vitro digestion model. The samples were selected based on the phytochemical profile and viable counts of *L. casei* 431^®^. The simulated model was prepared using a modified method described by Kim et al. [30]. In brief, the method involved successive steps of dissolving the powder in 10 mL of simulated saliva (Tris-HCl buffer, pH 7.7) containing 1 mg/mL α-amylase, followed by gastric digestion for 2 h and then intestinal digestion for another 2 h. An amount of 0.5 g of powder was dissolved in 10 mL of simulated saliva, which was then added to 20 mL of simulated gastric juice (SGS) composed of pepsin (1 mg/mL in 0.1 N HCl) at pH 2.0 and 1N HCl. 

After 2 h of gastric digestion at 37 °C, with continuous stirring at 150 rpm, a volume of 15 mL of the digested sample was transferred to simulated intestinal juice (SIS) consisting of pancreatin with sodium bicarbonate at pH 7.0, followed by incubating the samples for 2 h at 37 °C with continuous stirring at 150 rpm on an SI-300R orbital shaker (Medline Scientific, UK). The TFC of the sample was measured in the initial and final phases of each digestion stage (after the oral phase, before and after 2 h of gastric digestion, and before and after 2 h of intestinal digestion).

The bioavailability of flavonoids refers to the amounts of compounds released from the powder after gastrointestinal digestion that could become available for absorption into the systemic circulation [31]. Bioaccessibility was calculated using Equation (2):(2)$$Bioaccesibility\;\left(\%\right)=\frac{TFC_{1}}{TFC_{0}}\times 100$$
where *TFC*_1_ is the total flavonoid content (mg/g DW) in samples digested after gastrointestinal digestion, and *TFC*_0_ is the total flavonoid content (mg/g DW) in powder before gastrointestinal digestion.

For the survival rate of probiotics, the inoculated powder was dissolved in 10 mL of simulated saliva, and the aforementioned protocol describing the in vitro digestion of flavonoids was applied. In the initial step and after 2 h of gastric digestion or at 2 h of intestinal digestion, samples were analyzed in terms of the viability of probiotic cells. The survival rate of probiotic cells was calculated using Equation (3):(3)$$Survival\;rate\;\left(\%\right)=\frac{Log\;\left(N_{1}\right)}{Log\;\left(N_{0}\right)}\times 100$$
where *N*_1_ is the total number of viable cells after each stage of simulated digestion, and *N*_0_ is the initial total number of viable cells before exposure at each stage of digestion.

### 2.11. CIEL*a*b* Analysis of Freeze-Dried Powders

For the color analysis of powders, a CR 410 Chroma Meter (Konica Minolta, Tokyo, Japan) colorimeter was used to determine the color coordinates: L* (illumination, brightness, 0 black, 100 white), a* (positive value red, negative value green), and b* (positive value yellow, negative value blue). 

### 2.12. Powder Structure and Morphology

The powder morphology was observed by scanning electron microscopy (SEM). In order to assess the structural and morphological characteristics, each powder was placed on a sample holder and fixed with double-sided tape. Then, it was sputter-coated with gold and observed in a Quanta FEG 250 SEM (FEI, United States of America).

### 2.13. Food Formulation

Based on the phytochemical profile and cell viability, two powders were selected to formulate protein bars (KP and KPB). Then, the selected powders were added in a ratio of 6% to a formulation containing 25.0 g of nuts, 25.0 g of dates, 50.0 g of dried prunes, 25.0 g of rice flour, and 10 g of wheat germ. Prior to powder addition, the abovementioned ingredients were crushed and mixed well for 5 min, followed by powder addition and mixing to the extent that ensured the uniform distribution of the powders in the mixtures. The samples were molded into spherical shapes and stored at 4 °C for 24 h. Three variants were obtained, coded as V1 (with 6% addition of KPB), V2 (with 6% addition of KP), and control (C) with no powder added. The corresponding variants of protein bars were analyzed for total polyphenols, total flavonoids, carotenoid content, antioxidant activity, and *L. casei* 431^®^ cell counts.

### 2.14. Statistical Analyses

Analyses were performed in triplicate, and the results were expressed as mean and standard deviation (SD). Experimental data were subjected to one-way analysis of variance (ANOVA) after checking the normality and homoscedasticity to find significant differences. For post hoc analysis, the Tukey technique with a 95% confidence interval was used; *p* < 0.05 was deemed statistically significant. Minitab 18 software was used to perform the statistical analysis.

## 3. Results and Discussion

### 3.1. The Phytochemical Profiles of the Kiwi Extracts

In order to have a comprehensive view of kiwi byproducts’ phytochemical content, selected solvents were used in order to obtain extracts enriched in hydrophilic and lipophilic compounds, such as polyphenols, flavonoids, and carotenoids. The phytochemical profile and antioxidant activity are shown in Table 1.

As expected, the kiwi peels showed higher contents of polyphenols, flavonoids, and carotenoids, leading to significantly higher antioxidant activity. Our results support the hypothesis of the accumulation of bioactives in the outer layer of fruits. Wang et al. [32] suggested that most phenolic compounds, such as protocatechuic acid, chlorogenic acid, caffeic acid, rutin, p-hydroxybenzoic acid, and quercitin, are found in different parts of kiwifruit, with higher contents in the peels. These authors reported higher concentrations of flavonoids and polyphenols in kiwi peels, up to 8.13 mg gallic acid/g, compared with values ranging from 2.89 to 6.91 mg gallic acid/g in kiwi pulp. 

As expected, the carotenoid content had the same trend, with β-carotene and lycopene contents of 4.23 ± 0.027 mg/g DW and 1.06 ± 0.027 mg/g DW, respectively, in kiwi peels. Our results are consistent with the hypothesis that the carotenoid content in kiwifruits is due to the cumulative content of β-carotene and chlorophylls [33]. As regards antioxidant activity, the kiwi peel extracts showed higher values for both DDPH and ABTS radical scavenging activities, which were correlated with the higher contents of polyphenols and carotenoids. Antioxidant activity is often associated with the prevention or inhibition of processes that lead to cellular degradation, caused mainly by the effects of free radicals. Using multivariate correlation analysis, Zhang et al. [34] established that different groups of polyphenolic compounds are mainly responsible for antioxidant activity.

### 3.2. The Phytochemical Profiles and Cell Viability of Inoculated Freeze-Dried Kiwi Powders

In order to valorize the kiwi pomace and peels as added-value ingredients, the fresh materials were enhanced with different flours in order to improve additional properties, such as color, prebiotic activity, enhanced antioxidant activity, etc. Therefore, four variants of inoculated kiwi-based powders containing *L. casei* 431^®^ were obtained. The phytochemical profiles of the four variants are given in Table 2. 

It is well known that fruits and vegetables are rich in polysaccharides. Kiwifruits contain large amounts of pectin and dietary fiber, which have been found to improve the immune system and chronic diseases associated with constipation. Therefore, it is fair to consider that kiwifruits confer prebiotic effects on the intestinal microbiota [35]. These authors suggested that kiwifruit can act as a prebiotic in the selective intensification of the growth of lactic acid bacteria (*Lactobacillus* and *Bifidobacterium*) while inhibiting *Clostridium* and *Bacteriodes* ssp.

Table 2 gives the number of colony-forming units in each sample before and after freeze-drying. The variant based on kiwi pomace (KPO) showed an initial cell concentration of 10.63 log CFU/g DW, while after freeze-drying, a decrease to 7.89 log CFU/g DW was observed. In the case of the variant based on kiwi peels (KP), the initial probiotic load (10.64 log CFU/g DW) slightly decreased to 9.46 log CFU/g DW, whereas for the variants with black rice and buckwheat flour addition, a prebiotic effect was observed, with a higher value for the KPB variant. Although the freeze-drying process is considered a friendly preservation method with a low rate of biologically active compound degradation, a decrease in the viable counts of *L. casei* 431^®^ was observed to different extents in all samples. It can be inferred that the higher survival rate observed in samples based on peels with flour addition is due to the prebiotic effects of the fiber, vitamins, and bioactives. For example, Xie et al. [36] suggested a link between polyphenolic compounds in kiwi byproducts and the viable counts of lactic acid bacteria cells. The survival rate of lactic acid bacteria ranged from 73% to 88%. Based on these results, it is fair to attribute a probiotic character to all of the kiwi-based variants, given the suggested minimum concentration of viable cells of log 6 CFU/g.

The polyphenolic content of the samples was highly influenced by the particular formulation. The kiwi byproduct-based formulations showed a polyphenolic content varying from 10.56 ± 0.30 mg AGE/g DW to 13.16 ± 0.33 mg AGE/g, whereas the flavonoid content was influenced by the type of flour added (Table 2). No significant differences were found in antioxidant activity values. Table 2 shows the carotenoid contents of the samples, highlighting the higher contents of β-carotene and lycopene in the KPO sample (0.90 ± 0.07 mg/g DW and 0.57 ± 0.05 mg/g DW, respectively).

### 3.3. Color Parameters

The color of the kiwi-based powders was evaluated by CIEL*a*b* color parameters, including brightness L* (0 black/100 white), a* (red/green), and b* (yellow/blue) (Table 3).

The L* values differed significantly among freeze-dried variants and were affected by flour addition. Therefore, the brightest sample was the kiwi pomace-based variant (KPO), followed by the sample with buckwheat flour addition (KPB). The variants based on kiwi peels and kiwi pomace with black rice addition showed the darkest shade. The a* parameters had positive values in all samples, with the KPR variant closest to red, correlated with anthocyanin content. The kiwi peel-based variant showed an a* value close to 0, whereas the b* parameter had a shade close to yellow-green. These values of the color parameters are linked to the predominant bioactive in each powder. For example, the predominance of chlorophylls and carotenoids in the KPO variant, anthocyanin in KPR, and flavonoids in KP and KPB can be observed visually. 

### 3.4. Survival of L. Casei 431^®^ during Exposure to Simulated Gastrointestinal Conditions

Probiotics are microorganisms that, when administered in appropriate amounts, confer a health benefit to the host [37]. It is well known that probiotic survival in the gastrointestinal tract highly depends on the host, the presence of antibiotics, and interactions with different food constituents [38]. Probiotics are sensitive to temperature, pH variation, oxygen, bile acids, and salts [39]. Therefore, in order to assess the survival rate of *L. casei* 431^®^ in simulated gastrointestinal conditions, the powder with the highest microbial load was tested for the cell survival rate in simulated gastrointestinal conditions, that is, kiwi pomace with buckwheat addition. During oral digestion, a decrease in viable counts from 9.27 log CFU/g DW to 8.46 log CFU/g DW was found. After 2 h of gastric digestion followed by 2 h of intestinal digestion, the survival rate of *L. casei* 431^®^ was 80% and 48%, respectively.

### 3.5. In Vitro Bioaccessibility of Flavonoids

Flavonoids are a subgroup of polyphenols known for their high antioxidant activity, but they have low stability due to their molecular structure characteristics. Flavonoids are sensitive to factors such as temperature, light, pH, oxygen, enzymatic hydrolysis, etc. In order to evaluate the bioaccessibility of flavonoids from the KPB sample, simulated in vitro digestion was performed. The results showed an increase in total flavonoid content from the oral (1.70 ± 0.19 mg CE/g DW) to gastric environment (2.84 ± 0.14 CE/g DW), suggesting the release of flavonoids from the freeze-dried matrices. The controlled release of the flavonoids continued in the intestinal environment, reaching up to 4.61 ± 0.07 mg CE/g DW. The complex composition of the selected variant, rich in flavonoids and fiber, caused favorable conditions for controlled release during digestion. Xie et al. [36] reported a decrease in polyphenols and flavonoids from kiwi pomace during digestion. These authors reported a total polyphenol content ranging from 1.10 ± 0.17 mg AGE/100 g plant material in the oral cavity to 0.17 ± 0.02 mg AGE/100 g plant material in the intestine. The flavonoid content decreased from 4.68 ± 0.23 mg CE/100 g plant material in the oral environment to 0.75 ± 0.06 mg CE/100 g plant material in the intestine.

### 3.6. Microscopic Structure of the Powders

The microstructure of the inoculated powders highlights potentially important physical features and potential controlled release of the bacterial load and bioactives. Figure 1 shows the scanning electron microscopy (SEM) images used to visualize the network architectures formed in the structurally complex freeze-dried matrices. In the case of freeze-dried samples based on kiwi pomace and *L. casei* (Figure 1a) (KP), thin sheets with apparent flat faces showing corrugated surfaces can be observed. The structure shows some vesicular formations with regular shapes. Similar views can be observed in the case of the KPB variant (Figure 1b), with a more complex structure involving the presence of numerous vesicular formations with sizes of few micrometers. The variant with black rice addition (Figure 1c) shows a more complex aspect on the surfaces, with more polyhedral formations. The KPO variant has a non-uniform appearance, similar to KP, with smaller areas containing both vesicular and polyhedral formations distributed on curved surfaces connected by ridge areas (Figure 1d).

### 3.7. Food Formulations

The selected powders were added to a formula of protein bars at a ratio of 6%. Three product variants were obtained by adding KPB and KP powders to the protein bars and using a control sample without any powder added. The protein bars were tested for phytochemicals, antioxidant activity, and cell viability. The addition of kiwi peel powder led to a statistically different phytochemical profile of the protein bar (Table 4), with a higher flavonoid content (1.48 ± 0.01 mg CE/g DW), polyphenol content (4.76 ± 0.07 mg AGE/g DW), and antioxidant activity (23.75 ± 0.16 mM Trolox/g DW) when compared with the control sample. 

The carotenoid content increased compared to the control sample, with values for β-carotene of 0.72 ± 0.02 and 0.43 ± 0.06 mg/g DW, respectively. The phenolic compound behavior is strongly dependent on food composition. Kiwifruit is a good source of various bioactive compounds and therefore has a high level of antioxidant activity; therefore, it can be introduced into foods to increase their functional value. Tylewicz et al. [40] developed snack formulas with the addition of kiwifruit and proved that kiwi sticks had higher contents of total phenolic compounds, vitamin C, and flavonoids. The viable *L. casei* 431^®^ counts reached 9.36 log CFU/g DW in the sample with KPB addition and 9.95 log CFU/g DW in the sample with KP addition, highlighting the high functionality of both added-value foods.

*L. casei* 431^®^ is one of the most widely used probiotic cultures in the entire industry. It has also been found to produce many bioactive metabolites that can confer benefits to the host when consumed [41]. There is great potential in the field of novel functional foods and pharmacobiotics derived from the genus *Lacticaseibacillus* [42]. As the mechanisms behind their health-promoting capabilities are revealed, possible applications for these strains are being developed in the food, biotechnology, and medical fields [43].

## 4. Conclusions

Nowadays, consumers are more interested in sustainable food production due to the growing awareness of environmental pollution and the large amount of waste generated during conventional food processing. Moreover, consumers are more oriented toward the healthy aspects of food. Given that the food industry is facing the handling of thousands of tons of kiwi byproducts discarded each year, it is necessary to consider kiwi byproducts as good sources of functional ingredients. The present study is based on the formulation of multifunctional bio-ingredients based on kiwifruit pomace and peels, various flours, and cultures of lactic acid bacteria. These powders were obtained by freeze-drying and characterized in terms of phytochemical content, bioaccessibility of flavonoids after digestion, color, morphology, and structure, as well as the viability of *Lacticaseibacillus casei* 431^®^ cells. In order to test the potential for food functionalization, the powder was added to protein bars, and the products were subsequently characterized in terms of added value. The results confirm that kiwi pomace has the potential to be used as a functional food ingredient with a beneficial effect on consumer health and a prebiotic effect on lactic acid bacteria. The results are also promising for the utilization of natural antioxidants in combination with lactic acid bacteria in order to develop multifunctional ingredients for food, pharmaceutical, and cosmetic applications.

## Figures and Tables

**Figure 1 foods-11-02334-f001:**
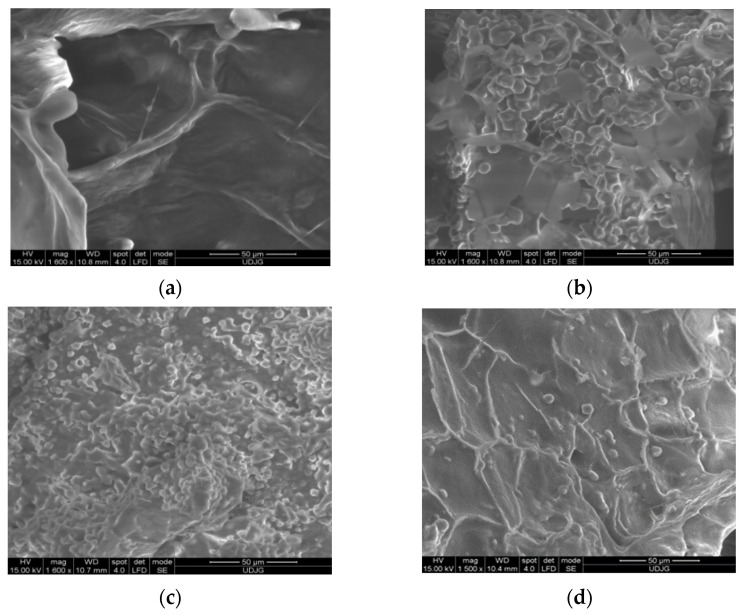
Structural and morphological characteristics of freeze-dried kiwi byproduct-based powders inoculated with *Lacticaseibacillus casei* 431^®^ (freeze-dried samples based on: (**a**) kiwi pomace; (**b**)—kiwi pomace and buckwheat flour; (**c**) kiwi pomace and black rice flour and (**d**)—kiwi peels.

**Table 1 foods-11-02334-t001:** Phytochemical characterization and antioxidant activity of kiwifruit pomace and peels.

Phytochemicals	Pomace	Peels
**Hydrophilic profile**		
Total flavonoid content, mg CE/g DW	1.97 ± 0.04 ^b^	5.34 ± 0.39 ^a^
Total polyphenolic content, mg GAE/g DW	3.79 ± 0.15 ^b^	9.71 ± 0.28 ^a^
Antioxidant activity, µM Trolox/g DW	9.68 ± 0.43 ^b^	51.09 ± 2.57 ^a^
**Hydrophobic profile**		
β-Carotene, mg/g DW	1.19 ± 0.065 ^b^	4.23 ± 0.027 ^a^
Lycopene, mg g DW	0.55 ± 0.042 ^b^	1.06 ± 0.027 ^a^
Total carotenoids, mg/g DW	1.55 ± 0.093 ^b^	5.70 ± 0.033 ^a^
Antioxidant activity, µM Trolox/g DW	5.92 ± 0.13 ^b^	8.62 ± 0.09 ^a^

Means in the same row that do not share a letter (a,b) are significantly different based on Tukey’s method with 95% confidence.

**Table 2 foods-11-02334-t002:** Phytochemical characterization of freeze-dried samples.

Variants	KPO	KPB	KBR	KP
	**Hydrophilic profile**			
Total polyphenolic content, mg GAE/g DW	12.46 ± 0.34 ^a^	10.56 ± 0.30 ^c^	11.51 ± 0.16 ^b^	13.16 ± 0.33 ^a^
Total flavonoid content, mg CE/g DW	1.49 ± 0.10 ^b^	2.03 ± 0.29 ^a^	2.17 ± 0.07 ^a^	1.89 ± 0.03 ^a,b^
Antioxidant activity, µM Trolox/g DW	57.67 ± 0.14 ^a^	56.32 ± 1.04 ^a^	56.93 ± 0.22 ^a^	56.91 ± 0.29 ^a^
	**Hydrophobic profile**			
β-Carotene, mg/g DW	0.90 ± 0.07 ^b^	0.84 ± 0.05 ^b^	0.81 ± 0.03 ^b^	1.47 ± 0.05 ^a^
Lycopene, mg/g DW	0.57 ± 0.05 ^a^	0.55 ± 0.01 ^a^	0.56 ± 0.07 ^a^	0.57 ± 0.04 ^a^
Total carotenoids, mg/g DW	1.04 ± 0.02 ^b^	1.04 ± 0.06 ^b^	1.04 ± 0.03 ^b^	1.89 ± 0.05 ^a^
AA, µM Trolox/g DW	3.90 ± 0.08 ^a^	3.99 ± 0.08 ^a^	4.07 ± 0.04 ^a^	4.06 ± 0.11 ^a^
*L. casei* 431^®^, log CFU/g DW	7.89 ^d^	9.27 ^b^	8.86 ^c^	9.46 ^a^

KPO—kiwi pomace with *L. casei* 431^®^; KPB—kiwi pomace with buckwheat flour and *L. casei* 431^®^; KBR—kiwi pomace with black rice flour and *L. casei* 431^®^; KP—kiwi peels and *L. casei* 431^®^. Means in the same row that do not share a letter (a–d) are significantly different based on Tukey’s method with 95% confidence.

**Table 3 foods-11-02334-t003:** Color parameters of freeze-dried powders.

Parameter	KPO	KPB	KBR	KP
**L***	39.48 ± 0.30 ^a^	37.76 ± 0.30 ^b^	13.63 ± 0.01 ^d^	33.78 ± 0.31 ^c^
**a***	7.22 ± 0.10 ^b^	6.055 ± 0.40 ^c^	13.67 ± 0.09 ^a^	0.605 ± 0.04 ^d^
**b***	22.32 ± 0.20 ^b^	13.67 ± 0.09 ^c^	0.91 ± 0.01 ^d^	25.43 ± 0.21 ^a^

L*—brightness, a*—redness/greenness, b*—yellowness/ blueness, KPO—kiwi pomace with *L. casei*; KPB—kiwi pomace with buckwheat flour and *L. casei*; KBR—kiwi pomace with black rice flour and *L. casei*; KP—kiwi peels and *L. casei* 431^®^. Means in the same row that do not share a letter (a–d) are significantly different based on Tukey’s method with 95% confidence.

**Table 4 foods-11-02334-t004:** Phytochemical profile and antioxidant activity of protein bars with freeze-dried powders addition.

Variants	C	V1	V2
Total polyphenolic content, mg GAE/g DW	3.95 ± 0.12 ^b^	4.18 ± 0.17 ^b^	4.76 ± 0.07 ^a^
Total flavonoid content, mg CE/g DW	1.36 ± 0.01 ^b^	1.37 ± 0.03 ^b^	1.48 ± 0.01 ^a^
Antioxidant activity, µM Trolox/g DW	22.51 ± 0.18 ^b^	22.67 ± 0.25 ^b^	23.75 ± 0.16 ^a^
β-caroten, mg/g DW	0.37 ± 0.02 ^b^	0.43 ± 0.02 ^b^	0.72 ± 0.06 ^a^
Lycopen, mg/g DW	0.17 ± 0.01 ^b^	0.19 ± 0.01 ^b^	0.37 ± 0.03 ^a^
Total carotenoids, mg/g DW	0.37 ± 0.02 ^c^	0.52 ± 0.02 ^b^	0.84± 0.06 ^a^

C—control protein bar with no powder added, V1—protein bars with 6% addition of freeze-dried powder based on kiwi-pomace, buckwheat and *Lacticaseibacillus casei* 431^®^, V2—protein bars with 6% addition of freeze-dried powder based on kiwi peels *Lacticaseibacillus casei* 431^®^. Means that on the same row do not share a letter (a,b,c) are significantly different, based on Tukey method and 95% confidence.

## Data Availability

Data is contained within the article.

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
