# Peer review of "Sustainable Design of Innovative Kiwi Byproducts-Based Ingredients Containing Probiotics"

_foods, 2022, doi:10.3390/foods11152334_

Round 1

Reviewer 1 Report

The manuscript describes a study on kiwifruit by-products used to develop food ingredients with added probiotics. The by-products were divided into skins and marc and the phytochemical profile was analyzed, determining polyphenols, flavonoids and carotenoids. The article is clear and well articulated, complete in the exposition of all the contents. The article requires only minor revisions:

1) add the keyword list in line 30

2) increase bibliographic references regarding introduction, materials and methods and where possible discussion of results..

3) increase the bibliographic references which are rather scarce considering the length and completeness of the article: line 43: Insert biliographic references on the health aspects of kiwi  fruit line 56: insert some bibliographic references to to which it refers line 198. insert the bibliographic reference of the famous method used lines 252-254: insert the bibliographic references to which it refers lines 313-315: insert the bibliographic to which it refers

Author Response

Authors:

The authors would like to thank the reviewers for the close reading and for the proper suggestions and comments aimed at improving the paper. The present version of the paper has been revised. The changes performed in order to comply with the suggestions of the reviewer are given in different colors, according to reviewers comments.

REVIEWER 1: GENERAL COMMENTS

The manuscript describes a study on kiwifruit by-products used to develop food ingredients with added probiotics. The by-products were divided into skins and marc and the phytochemical profile was analyzed, determining polyphenols, flavonoids and carotenoids. The article is clear and well articulated, complete in the exposition of all the contents. The article requires only minor revisions.

Authors answer: The authors would like to thank the reviewers for the close reading and for the proper suggestions and comments aimed at improving the paper. The present version of the paper has been revised. The changes performed in order to comply with the suggestions of the reviewer are given in red, according to reviewer comments.

Comment 1: add the keyword list in line 30;

Authors answer: Thank you for your observation. Key words were added at lines 30-31, as follow:

Line 30-31: Keywords: kiwi by-products; probiotic; prebiotic; antioxidant activity; Lactobacillus casei; ingredients; functional foods.

Comment 2: increase bibliographic references regarding introduction, materials and methods and where possible discussion of results.

Authors answer: Thank you for your observation. Key words were added at line 54-88, as follow:

Lines 54-88: Significant data from the literature have shown that one of the most recommended way to prevent and treat gastrointestinal alterations is to alter the commensal microbiota [10, 11, 12]. It is well known that microorganisms coexist with eukaryotic cells at the mucosal surfaces of vertebrates in a complex and harmonious symbiosis [11]. However, administration of antibiotics at large scale cause various side effects, such as disrupting normal microflora in the human body, by destroying normal gut and genital tract bacteria [13]. These imbalances can be adjusted by probiotics supplementation in diet, respectively providing a load of live bacterial cells sufficient for beneficial effects on human health, both at metabolic and immune levels [11]. The engineering of probiotic foods is challenging due to their low acid-bile tolerances, the requirements of lack of pathogenicity and nontoxicity, surviving and metabolizing in the gut environment, resistant to low pH and organic acid and the potential of remaining viable for long period under storage and other conditions [14]. An optimal population of probiotic bacteria is critical for the preservation and proper functioning of the digestive system [15].

Prebiotics are defined as non-fermentable components to be transferred into the colon to be selectively used by host microorganisms [16]. Multiples benefits are suggested for prebiotics, such as to mediate host health such as to improve in intestinal function, regulation of glucose and lipid metabolism, immune response, bone health and regulation of satiety [17]. It has been suggested that prebiotics may be classified in three categories: oligosaccharides (fructooligosaccharide, xylooligosaccharide, galactooligosaccharide, isomaltose, inulin, etc.), fiber (β-glucan, pectin, cellulose, dextrin, etc.), and polyols (xylitol, mannitol, lactulose, etc.) [17], whereas other compounds have been proposed as candidate for prebiotics, such as linoleic acid, polyunsaturated fatty acids, phenols and polyphenols, such as anthocyanins [18].

Given the need to use prebiotics in engineering of probiotic foods, different complex food matrices should be considered, to include both fibers, polyphenolics and oligosaccharides. For example, black rice (Oryza sativa L.) flour is a valuable source of protein, fat, carbohydrates, phenols, flavonoids, and anthocyanins [19] and could be a potent candidate for complex prebiotic activity. Additionally, buckwheat is a type of underutilized pseudocereals, belonging to the genus Fagopyrum and it could be regarded as a potential source for food and nutritional applications [20]. According to FoodData Central [21], the proximate composition of buckwheat includes starch as the major component (~70%), followed by protein (~12%), dietary fibers (~10%), lipids (~3%) and ash (2.5%), whereas minor components with biological significance such as polyphenols, D-chiro-inositol and vitamins were also reported [22].

References:

  1. Kim, S.; Covington, A.; Pamer, E.G. The intestinal microbiota: antibiotics, colonization resistance, and enteric pathogens. Rev. 2017, 279 (1), 90–105.
  2. Yadav, A.; Chandra, H.; Maurya, V.K. Probiotics: recent advances and future prospects. Plant Dev. Sci. 2017, 9(11), 967–975.
  3. Song, H.Y.; Zhou, L.; Liu, D.Y.; Yao, X.J.; Li, Y. What roles do probiotics play in the eradication of Helicobacter pylori? Current knowledge and ongoing research. Res. Pract. 2018, 2018, 9379480.
  4. Tegegne, B.A.; Kebede, B. Probiotics, their prophylactic and therapeutic applications in human health development: A review of the literature. Heliyon, 2022, 8, e0972.
  5. Anosike, F.C.; Onyemah, C.O.; Ossai, C.U.; Ofoegbu, J.N.G.; Okpaga, F.O.; Ikpeama, C.C.; Nkwegu, F.M.; Nwankwo, S.C.; Onyeji, G.N.; Inyang, P.; Ndifon, E.M., Emeka, C.P.O. Probiotic potential and viability of bacteria in fermented African oil bean seed (Pentaclethra macropyhlla): A mini review. Applied Food Res. 2022, 2, 1000082.
  6. Likotrafiti, E.; Rhoades, J. Probiotics, Prebiotics, Synbiotics, and Foodborne Illness, In: Probiotics, Prebiotics, and Synbiotics. Bioactive Foods in Health Promotion, Edited by Watson, R.R. and Preedy, V.R., Academic Press. 2016, Pages 469-476.
  7. Wang, M.; Zhang, Z.; Sun, H.; He, S.; Liu, S.; Zhang, T.; Wang, L.; Ma, G. Research progress of anthocyanin prebiotic activity: A review. Phytomed. 2022, 102, 154145
  8. Gibson, G. R., Hutkins, R., Sanders, M. E., Prescott, S. L., Reimer, R. A., Salminen, S. J., et al. Expert consensus document: The International Scientific Association for Probiotics and Prebiotics (ISAPP) consensus statement on the definition and scope of prebiotics. Nature Reviews. Gastroenterol. Hepatol. 2017, 14(8), 491–502.
  9. Alves-Santos, A.M.; Araújo Sugizaki, C.S.; Lima, G.C.; Veloso Naves, M.M. Prebiotic effect of dietary polyphenols: A systematic review. Funct. Foods. 2020, 74, 104169.
  10. Shen, Y.; Jin, L.; Xiao, P.; Lu, Y., Bao, J. S. Total phenolics, flavonoids, antioxidant capacity in rice grain and their relations to grain color, size and weight. Cereal Sci. 2009, 49(1), 106–111.
  11. Zhu, F. Buckwheat proteins and peptides: Biological functions and food applications. Trends Food Sci. Technol. 2021, 110, 155–167.
  12. (2020). The food and agriculture organization corporate statistical database. http://www.fao.org/faostat/en/#data/QC Accessed on July the 29th, 2022.
  13. Zhu, F. Buckwheat starch: Structures, properties, and applications. Trends Food Sci. Technol. 2016, 49, 121–135.

Comment 3: increase the bibliographic references which are rather scarce considering the length and completeness of the article: line 43: Insert biliographic references on the health aspects of kiwi  fruit line 56: insert some bibliographic references to to which it refers line 198. insert the bibliographic reference of the famous method used lines 252-254: insert the bibliographic references to which it refers lines 313-315: insert the bibliographic to which it refers.

Authors answer: The references section, both in text and references was updated. Please see the revised form of the manuscript.

Reviewer 2 Report

The article “Sustainable design of innovative kiwi by-products based ingredients containing probiotics” focuses on an interesting and innovative research topic within the scope of Foods as well as of the SI “Valorization of Food Processing By-Products”. However, the authors must be aware of some corrections to be made in the article, as cited below:

- The keywords are lacking in the article.

- The reason to add prebiotic ingredients to the kiwi by-products powders must be clear in the Introduction section, as well as the selection of black rice flour and buckwheat flour as prebiotic ingredients. What is the reported evidence about the prebiotic properties of black rice flour and buckwheat flour?

- The code of the L. casei strain used in the experiments must be informed, as well as references for its probiotic properties. Is this strain already used as a probiotic? The authors must make it clear as a background to give the potential probiotic functionality to the designed products.

- Correct the name of the strain to “Lacticaseibacillus casei” followed by the identification code.

- Use “Viable counts of L. casei” instead of “Viability of L. casei”. Check it throughout the article.

- Authors must explain the rationale to examine only the KPB sample for bioaccessibility analysis. Could it be representative of the other samples?

- Use “Results and discussion” as a section instead of “Results”.  

- The use of the term “In vitro bioaccessibility of L. casei” sounds weird and uncommon in the literature. Authors should use “Survival of L. casei during exposure to simulated gastrointestinal conditions”.

Author Response

GENERAL COMMENTS: The article “Sustainable design of innovative kiwi by-products based ingredients containing probiotics” focuses on an interesting and innovative research topic within the scope of Foods as well as of the SI “Valorization of Food Processing By-Products”. However, the authors must be aware of some corrections to be made in the article, as cited below:

Authors answer: The authors would like to thank the reviewers for the close reading and for the proper suggestions and comments aimed at improving the paper. The present version of the paper has been revised. The changes performed in order to comply with the suggestions of the reviewer are given in blue, according to reviewer comments.

Comment 1: The keywords are lacking in the article.

Authors answer: Thank you for your observation. Key words were added at line 30-31, as follow:

Line 30-31: Keywords: kiwi by-products; probiotic; prebiotic; antioxidant activity; Lactobacillus casei; ingredients; functional foods.

Comment 2: The reason to add prebiotic ingredients to the kiwi by-products powders must be clear in the Introduction section, as well as the selection of black rice flour and buckwheat flour as prebiotic ingredients. What is the reported evidence about the prebiotic properties of black rice flour and buckwheat flour?

Authors answer: Thank you for your suggestion. Additional information was added, as follow:

Lines 68-88: Prebiotics are defined as non-fermentable components to be transferred into the colon to be selectively used by host microorganisms [16]. Multiples benefits are suggested for prebiotics, such as to mediate host health such as to improve in intestinal function, regulation of glucose and lipid metabolism, immune response, bone health and regulation of satiety [17]. It has been suggested that prebiotics may be classified in three categories: oligosaccharides (fructooligosaccharide, xylooligosaccharide, galactooligosaccharide, isomaltose, inulin, etc.), fiber (β-glucan, pectin, cellulose, dextrin, etc.), and polyols (xylitol, mannitol, lactulose, etc.) [17], whereas other compounds have been proposed as candidate for prebiotics, such as linoleic acid, polyunsaturated fatty acids, phenols and polyphenols, such as anthocyanins [18].

Given the need to use prebiotics in engineering of probiotic foods, different complex food matrices should be considered, to include both fibers, polyphenolics and oligosaccharides. For example, black rice (Oryza sativa L.) flour is a valuable source of protein, fat, carbohydrates, phenols, flavonoids, and anthocyanins [19] and could be a potent candidate for complex prebiotic activity. Additionally, buckwheat is a type of underutilized pseudocereals, belonging to the genus Fagopyrum and it could be regarded as a potential source for food and nutritional applications [20]. According to FoodData Central [21], the proximate composition of buckwheat includes starch as the major component (~70%), followed by protein (~12%), dietary fibers (~10%), lipids (~3%) and ash (2.5%), whereas minor components with biological significance such as polyphenols, D-chiro-inositol and vitamins were also reported [22].

References:

  1. Wang, M.; Zhang, Z.; Sun, H.; He, S.; Liu, S.; Zhang, T.; Wang, L.; Ma, G. Research progress of anthocyanin prebiotic activity: A review. Phytomed. 2022, 102, 154145
  2. Gibson, G. R., Hutkins, R., Sanders, M. E., Prescott, S. L., Reimer, R. A., Salminen, S. J., et al. Expert consensus document: The International Scientific Association for Probiotics and Prebiotics (ISAPP) consensus statement on the definition and scope of prebiotics. Nature Reviews. Gastroenterol. Hepatol. 2017, 14(8), 491–502.
  3. Alves-Santos, A.M.; Araújo Sugizaki, C.S.; Lima, G.C.; Veloso Naves, M.M. Prebiotic effect of dietary polyphenols: A systematic review. Funct. Foods. 2020, 74, 104169.
  4. Shen, Y.; Jin, L.; Xiao, P.; Lu, Y., Bao, J. S. Total phenolics, flavonoids, antioxidant capacity in rice grain and their relations to grain color, size and weight. Cereal Sci. 2009, 49(1), 106–111.
  5. Zhu, F. Buckwheat proteins and peptides: Biological functions and food applications. Trends Food Sci. Technol. 2021, 110, 155–167.
  6. (2020). The food and agriculture organization corporate statistical database. http://www.fao.org/faostat/en/#data/QC Accessed on July the 29th, 2022.
  7. Zhu, F. Buckwheat starch: Structures, properties, and applications. Trends Food Sci. Technol. 2016, 49, 121–135.

Comment 3: The code of the L. casei strain used in the experiments must be informed, as well as references for its probiotic properties. Is this strain already used as a probiotic? The authors must make it clear as a background to give the potential probiotic functionality to the designed products.

Authors answer: Thank you for your suggestion. Additional information was added at lines:

Lines 109-112: The probiotic character of L. casei 431® strain is described in several papers [23, 24] and has been associated with health benefits within several areas of health, including immune health, respiratory healthy and bowel function. 

References:

  1. Jespersen, L.; Tarnow, I.; Eskesen, D.; Melsaether Morberg, C.; Michelsen, B.; Bügel, S.;Dragsted, L.O.; Rijkers, G.T.; Calder, F.C. Effect of Lactobacillus paracasei subsp. paracasei, L. casei 431 on immune response to influenza vaccination and upper respiratory tract infections in healthy adult volunteers: a randomized, double-blind, placebo-controlled, parallel-group study.  J. Clin. Nutr2015, 101(6), 1188-96.
  2. Trachootham, D.; Chupeerach, C.; Tuntipopipat, S.; Pathomyok, L.; Boonnak, K., Praengam, K.; Promkam, C.; Santivarangkn, C. Drinking fermented milk containing Lactobacillus paracasei 431 (IMULUS™) improves immune response against H1N1 and cross-reactive H3N2 viruses after influenza vaccination: A pilot randomized triple-blinded placebo controlled trial. J. Funct. Foods. 2017, 33, 1–10. 

Comment 4: Correct the name of the strain to “Lacticaseibacillus casei” followed by the identification code.

Authors answer: Thank you for your suggestion. The name of the strain was corrected all over the manuscript. Please, see the revised form.

Comment 5: Use “Viable counts of L. casei” instead of “Viability of L. casei”. Check it throughout the article.

Authors answer: Thank you for your suggestion. Additional information was added, as follow:

Line 181: Viable counts of L. casei 431®

Comment 6: Authors must explain the rationale to examine only the KPB sample for bioaccessibility analysis. Could it be representative of the other samples?

Authors answer: Thank you for your suggestion. Additional information was added, as follow:

Lines 191-192: The samples was selected based on the phytochemical profile and viable counts of L. casei 431®.

Comment 7: Use “Results and discussion” as a section instead of “Results”.  

Authors answer: Thank you for your suggestion. Additional information was added, as follow:

Line 252: Results and discussion

Comment 8: The use of the term “In vitro bioaccessibility of L. casei” sounds weird and uncommon in the literature. Authors should use “Survival of L. casei during exposure to simulated gastrointestinal conditions”.

Authors answer: Thank you for your suggestion. Additional information was added, as follow:

Line 337: Survival of L. casei 431® during exposure to simulated gastrointestinal conditions

Round 2

Reviewer 2 Report

The authors have done good work with corrections to the articles according to the reviewers' comments.  The article can be accepted in its current form.